# SymAttack: Symmetry-aware Imperceptible Adversarial Attacks on 3D Point Clouds

## ABSTRACT

Adversarial attacks on point clouds are crucial for assessing and improving the adversarial robustness of 3D deep learning models. Despite leveraging various geometric constraints, current adversarial attack strategies often suffer from inadequate imperceptibility. Given that adversarial perturbations tend to disrupt the inherent symmetry in objects, we recognize this disruption as the primary cause of the lack of imperceptibility in these attacks. In this paper, we introduce a novel framework, symmetry-aware imperceptible adversarial attacks on 3D point clouds (SymAttack), to address this issue. Our approach starts by identifying part- and patch-level symmetry elements, and grouping points based on semantic and Euclidean distances, respectively. During the adversarial attack iterations, we intentionally adjust the perturbation vectors on symmetric points relative to their symmetry plane. By preserving symmetry within the attack process, SymAttack significantly enhances imperceptibility. Extensive experiments validate the effectiveness of SymAttack in generating imperceptible adversarial point clouds, demonstrating its superiority over the state-of-the-art methods. Codes will be made public upon paper acceptance.

## CCS CONCEPTS

• **Security and privacy** → **Software and application security**;
• **Computing methodologies** → **Computer vision**; **Shape representations**; **Shape analysis**.

## KEYWORDS

Adversarial attacks, Symmetry, 3D point clouds, Deep neural networks

## 1 INTRODUCTION

The rise of deep learning combined with the availability of affordable depth-sensing technologies has thrust the analysis of 3D point clouds through deep neural networks (DNNs) into the forefront of research [8]. Nevertheless, the progression in this field is challenged by the vulnerability of DNN classifiers to adversarial attacks, as highlighted in recent studies [16, 38]. These attacks introduce subtle, often imperceptible, modifications to point cloud inputs, leading DNN models to make incorrect predictions. Such vulnerabilities significantly hinder the real-world application of these technologies, particularly in safety-critical scenarios like autonomous driving [20]. Therefore, exploring adversarial attacks on point clouds is

Unpublished working draft. Not for distribution.

Permission to make digital or hard copies of all or part of this work for personal or classroom use is granted without fee provided that copies are not made or distributed for profit or commercial advantage and that copies bear this notice and the full citation on the first page. Copyrights for components of this work owned by others than the author(s) must be honored. Abstracting with credit is permitted. To copy otherwise, or republish, to post on servers or to redistribute to lists, requires prior specific permission and/or a fee. Request permissions from permissions@acm.org.
*MM '24, October 28–November 1, 2024, Melbourne, Australia*
© 2024 Copyright held by the owner/author(s). Publication rights licensed to ACM.
ACM ISBN 978-1-4503-XXXX-X/18/06
https://doi.org/XXXXXXX.XXXXXXX

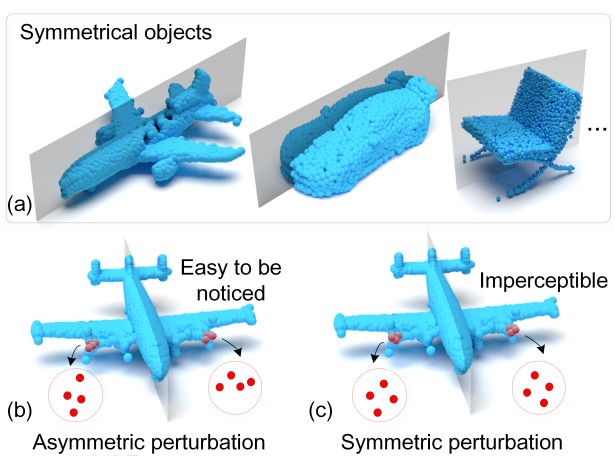

Figure 1: (a) Most 3D objects exhibit symmetry. (b) Traditional adversarial attacks compromise symmetry in 3D point clouds, making them easily noticeable. (c) By preserving this property, SymAttack achieves greater imperceptibility.

crucial for assessing and improving the adversarial robustness of 3D deep learning models against these security concerns.

Achieving imperceptibility is essential for the effective execution of adversarial attacks on 3D point clouds, leading to extensive research efforts in this domain. These techniques predominantly divide into two categories. The first approach involves introducing objects that are conspicuous yet contextually plausible and unlikely to arouse suspicion within the given scenario, such as a toy airplane close to the target object as in [38], or utilizes physics-based deformations [29] to integrate attacks seamlessly into human perception. The second, and more widely adopted, strategy aims to minimize alterations by applying a variety of constraints. Traditional approaches in this category utilize metrics such as the $l_2$-norm, Chamfer distance, and Hausdorff distance to quantify and reduce perturbations. Building on these, more recent innovations focus on preserving the original geometric features of 3D point clouds. This is achieved by ensuring geometric consistency [33] or by directing adjustments along an object's normal [15] or tangential planes [9]. Despite these advancements, the imperceptibility of adversarial attacks on 3D point clouds remains unsatisfactory, e.g., visible outliers are still present.

This raises a crucial question: despite the application of geometric constraints, why do adversarial attacks remain perceptible? Indeed, the effectiveness of imperceptibility hinges on the preservation of certain regularities provided by these constraints, aligning with the common understanding that the human eye excels at detecting inconsistencies within regular structures [31]. However, typical adversarial perturbations tend to disrupt a crucial regularity, i.e., the symmetrical property, as illustrated in Fig. 1. Therefore,

addressing the loss of symmetry in adversarial attacks is anticipated to enhance their imperceptibility.

In this paper, we propose a novel framework named symmetry-aware imperceptible adversarial attacks on 3D point clouds (SymAttack), designed to enhance imperceptibility by emphasizing the preservation of symmetry within the attack process. Our strategy commences with the delineation of symmetric elements in point clouds. This involves identifying part-level symmetric elements, distinguished by semantic distances, and patch-level symmetric elements, which aggregate points based on Euclidean distances, and then proceed to verification. During the adversarial attack iterations, we intentionally adjust the perturbation vectors on identified symmetric elements relative to their symmetry plane to conserve their natural symmetry. As a consequence, our method effectively minimizes the disruptions in symmetry within adversarial point clouds, thereby significantly enhancing their imperceptibility. We validate the effectiveness of our SymAttack framework in attacking four common DNN classifiers for 3D point clouds. Extensive experimental results show that the generated adversarial point clouds are significantly more imperceptible, outperforming those generated by state-of-the-art methods. Besides, we demonstrate that SymAttack is capable of resisting various adversarial defenses.

Overall, our contribution is summarized as follows:

- We are the first to attribute the inadequate imperceptibility of adversarial attacks on 3D point clouds to the large deviation of symmetry.
- We develop a novel adversarial attack framework that preserves point cloud symmetry by identifying patch- and part-level symmetry elements and then adjusting the perturbations on them.
- We show by experiments that our SymAttack framework with preserving symmetry achieves superior performance in terms of imperceptibility under various metrics.

## 2 RELATED WORK

### 2.1 Adversarial Attacks on 3D Point Clouds

Adversarial attacks, which originated in the domain of 2D image classification, have expanded to encompass 3D point clouds, introducing distinct challenges. These attacks can generally be categorized into three types: addition-based, where new points are introduced to induce misclassification [38]; deletion-based, involving the strategic removal of pivotal points to influence the model's decision [34, 43, 48]; and perturbation-based, which involves making subtle modifications to the coordinates of points to deceive the models [12, 38, 47]. This study focuses on exploring the intricacies of perturbation-based adversarial attacks.

Pioneering efforts in perturbation-based 3D adversarial attacks were spearheaded by Xiang et al. [38] and Liu et al. [16], who skillfully adapted the C&W [1] and FGSM [7] strategies for 3D contexts. Most subsequent work has built upon their foundations, introducing additional constraints to achieve specific goals, such as imperceptibility [15, 33]. Other approaches to adversarial attacks include the use of isometric transformations, primarily rotations, to manipulate point clouds [47], and limiting perturbations to a small subset of points instead of all points [12]. Furthermore, with the rise of generative adversarial networks (GANs) [6], Lee et al. [13] and

Zhou et al. [49] have also experimented with using GANs to generate adversarial point clouds. Recently, Tang et al. [29] introduced a method from a manifold perspective, involving parameter plane stretching that affects the underlying 2-manifold surface for a more effective attack. In this paper, we focus primarily on adversarial attack methods capable of achieving high imperceptibility.

### 2.2 Imperceptible 3D Adversarial Attacks

To achieve imperceptibility in adversarial attacks on point clouds, prevalent strategies employ constraints such as the $l_2$-norm, Chamfer distance, and Hausdorff distance to reduce the discrepancies between the original and modified point clouds [16, 38, 49]. GeoA$^3$ [33] has enhanced these methods by incorporating advanced geometric regularity constraints, leading to improved outcomes. Subsequent research has explored the use of directional perturbations, with the aim of aligning them with each point's normal vector [15] or along tangential planes [9], to preserve the natural shape of objects. More recently, Tang et al. [28] sought to improve imperceptibility from a manifold perspective by minimizing distortion. Despite its effectiveness, this approach relies on the creation of manifold mappings, which presents significant challenges for point clouds with complex shapes. In contrast to these approaches, our work highlights the importance of symmetry preservation—an essential but often neglected aspect in ensuring the imperceptibility of attacks.

### 2.3 Deep 3D Point Cloud Classification

Significant advancements have been made in deep learning models for 3D point cloud classification. Voxel-based methods transform 3D spaces into voxel grids for analysis with 3D CNNs, as detailed in studies using binary grids and sophisticated voxel representations to classify spaces [17]. While these methods have seen improvements, their performance is limited by the need for high resolution. In contrast, the introduction of PointNet [22] represented a paradigm shift, advocating for the direct processing of point clouds. This innovation has spurred subsequent developments, including the utilization of hierarchical structures [23], point-specific convolutions [14, 30, 35, 39], and graph-based CNNs [2, 25, 26, 32, 45], which have continually raised performance standards. For a comprehensive review of the state of the art in deep learning models for 3D point cloud classification, please refer to the survey papers [8, 10]. In this paper, we aim to attack these DNN classifiers in an imperceptible manner.

### 2.4 Symmetry for 3D Shape Processing

In the field of 3D shape processing, symmetry stands as a pivotal aspect influencing various research domains [19]. It plays a critical role in object detection, significantly contributing to the accurate determination of object's 6D poses [46], and facilitates the segmentation process, enabling the division of point clouds into distinct and meaningful segments [4, 42]. Additionally, the integration of symmetry in shape matching substantially improves alignment accuracy [11, 41]. In this paper, we attribute the lack of imperceptibility in adversarial attacks on 3D point clouds to the disruptions in symmetry, and we seek to preserve symmetry while applying adversarial perturbations to enhance imperceptibility.

**Figure 2: Illustration of our SymAttack framework: given a symmetric point cloud as input, the approach first identifies part- and patch-level symmetric elements and then iteratively generates and adjusts initially asymmetric perturbations on points within symmetric elements to maintain symmetry.**

## 3 PROBLEM FORMULATION

### 3.1 Typical Adversarial Attacks

Given a point cloud $\mathcal{P}$ in $\mathbb{R}^{n \times 3}$ with its corresponding label $y$ from the set $\{1, ..., C\}$, where $C$ is the number of categories, the goal of perturbation-based adversarial attacks is to fool a 3D deep classification model $\mathcal{F}$ into making incorrect predictions. This is accomplished by crafting an adversarial point cloud $\mathcal{P}^{adv}$ through the application of imperceptible perturbations. The generation of the adversarial point cloud is formally defined as:

$$P_i^{adv} = P_i + \sigma_{P_i} \cdot \overrightarrow{d_{P_i}}, \tag{1}$$

where $P_i$ is the $i$-th point in $\mathcal{P}$, $\sigma_{P_i}$ denotes the magnitude of perturbation on $P_i$, and $\overrightarrow{d_{P_i}}$ is the unit vector direction of the perturbation.

The process to determine the perturbation $\sigma_{P_i} \cdot \overrightarrow{d_{P_i}}$ involves solving the optimization problem below, typically using gradient descent methods:

$$\min L_{mis}(\mathcal{F}, \mathcal{P}^{adv}, y) + \lambda_1 D(\mathcal{P}, \mathcal{P}^{adv}), \tag{2}$$

where $L_{mis}(\cdot, \cdot, \cdot)$ is the loss function aimed at inducing misclassification, e.g., the negation of cross-entropy loss, $D(\cdot, \cdot)$ represents the distortion constraints to maintain imperceptibility, and $\lambda_1$ is a balancing parameter. Here, our primary emphasis is on untargeted attacks, and the framework can be easily adapted to accommodate targeted attacks as well.

**Discussion.** Traditional strategies for ensuring imperceptibility in adversarial attacks typically involve the application of geometric constraints, such as the $l_2$-norm, Chamfer distance, Hausdorff distance, and curvature, to regulate perturbations. However, these approaches often neglect a critical aspect—symmetry. The human visual system is highly sensitive to deviations in symmetry, emphasizing the necessity of preserving this feature to secure the imperceptibility of adversarial point clouds.

### 3.2 Symmetry-aware Adversarial Attacks

Symmetry is a fundamental attribute of geometric shapes, often observed when one half of an object serves as a mirror reflection of the other half. Suppose point $P_i$ and its counterpart $\hat{P}_i$ are symmetric with respect to the symmetry plane $\mathcal{S}$, *symmetry-aware adversarial attacks* require that the adversarial perturbation vectors applied to $P_i$ and $\hat{P}_i$ must also be symmetric relative to $\mathcal{S}$. Formally, this

requirement can be expressed as:

$$Mirror(\sigma_{P_i} \cdot \overrightarrow{d_{P_i}}, \mathcal{S}) = \sigma_{\hat{P}_i} \cdot \overrightarrow{d_{\hat{P}_i}}, \tag{3}$$

where $Mirror(\sigma_{P_i} \cdot \overrightarrow{d_{P_i}}, \mathcal{S})$ denotes the perturbation vector mirrored across the symmetry plane $\mathcal{S}$.

By maintaining the symmetry of perturbations on two symmetric points within 3D shapes, such adversarial attacks preserve the original shape's symmetry, thereby achieving a superior level of imperceptibility compared to conventional methods.

## 4 METHOD

In this section, we detail the process of detecting symmetric elements within a point cloud and subsequently introduce the symmetry-aware imperceptible adversarial attacks framework (SymAttack). Please refer to Fig. 2 for an illustration.

### 4.1 Symmetric Element Detection

Given that the requirement for adversarial attacks to maintain symmetry across the entire shape is overly stringent, we opt for selecting a subset of representative symmetric elements. We begin by identifying the symmetry plane, proceed by sampling symmetric elements, and finally verify them.

*4.1.1 Identify Symmetry Plane.* The symmetry plane in an object is an imaginary plane that divides the object into two equal parts, which are mirror images of each other with respect to this plane. Ideally, if you flip the object along its symmetry plane, the two sides will align perfectly. A classic example is the exterior of most cars, where the left side mirrors the right side across the vehicle's longitudinal plane, as illustrated in Fig. 2.

To identify the symmetry plane $\mathcal{S}$, we utilize the octree representation combined with the principal axis transform to accurately assess the degree of symmetry in objects, as described in [18]. It is possible for an object to have multiple planes of symmetry; our analysis focuses on the plane demonstrating the highest degree of symmetry. Additionally, a minimal value in this assessment suggests the lack of a symmetry plane within the object.

*4.1.2 Sampling Symmetric Elements on One Side.* To facilitate the measurement and enforcement of symmetry, we concentrate on symmetric elements, i.e., points, parts, or components that, when aligned relative to a symmetry plane, display mirrored counterparts.

For each element $E_i$ and its counterpart $\hat{E}_i$, this principle can be formally defined as:

$$Mirror(E_i, \mathcal{S}) = \hat{E}_i. \tag{4}$$

An exemplary case is observed in automobiles, where features such as wheels, windows, and doors are symmetrically aligned across the vehicle's central symmetry plane.

To capture symmetric elements, we initially identify part-level elements with semantic features, such as the wheels of a car. Subsequently, we sample patch-level elements from the remaining point cloud to ensure comprehensive coverage of the entire shape. For convenience, we conduct sampling only on one side of the shape. **Sampling Part-level Elements.** We start with farthest point sampling to select key points $\{Q_i\}_{i=1:K}$. Utilizing these points, we proceed to create part-level elements by employing flooding, guided by the criterion $E_i^1 = \{P | d_S(f(P), f(Q_i)) < \tau_s\}$, where $d_S(\cdot, \cdot)$ measures the semantic distance, $f(\cdot)$ is a function implemented by the pretrained auto-encoder, e.g., Point-M2AE [44], to extract semantic features, and $\tau_s$ is a threshold. It's noteworthy that multiple key points can converge into a single element. For efficiency, we specifically exclude elements containing more than $\tau_P$ points, yielding the final collection of part-level elements $\mathcal{E}^1$.
**Sampling Patch-level Elements.** To ensure that elements cover the entire surface of the object, in regions not occupied by part-level elements, we localize patch-level elements in areas surrounding $Q_i$ as center points. This is achieved through the criterion $E_i^2 = \{P | d_E(P, Q_i) < \tau_r\}$, where $d_E(\cdot, \cdot)$ measures the Euclidean distance, and $\tau_r$ is a predefined threshold. This procedure results in the final collection of patch-level elements, denoted as $\mathcal{E}^2$.

*4.1.3 Symmetry Verification.* Given elements on one side, we proceed to identify their corresponding counterparts on the opposite side and verify their symmetry. Specifically, for any given $P \in E$, where $E \in \mathcal{E}^1 \cup \mathcal{E}^2$, we locate its counterpart on the opposite side as the point closest to $Mirror(P, \mathcal{S})$, thereby forming $E'$ symmetrical to $E$. To verify their symmetry, we measure the $l_2$ distance between element $E$ and its mirror $E'$ relative to the symmetry plane. Elements with an average distance greater than $\tau_v$ are filtered out.

## 4.2 Symmetry-aware Imperceptible Adversarial Attacks

During the adversarial attack process, we aim to maintain the object's inherent symmetry; that is, if a symmetric element on the object undergoes an adversarial perturbation, its mirrored counterpart receives a symmetrically equivalent perturbation. This method, named symmetry-aware imperceptible adversarial attacks (SymAttack), comprises two main iterative steps: generating initial perturbations, and symmetry-aware adjustment, as illustrated in Fig. 2.
**Generating Initial Perturbations.** We simply employ IFGM [3] to generate initial perturbations. It is noteworthy that alternative methods could also be employed.
**Symmetry-aware Adjustment.** Given a point $P_i \in E$ with its adversarial perturbation $\sigma_{P_i} \cdot \overrightarrow{d_{P_i}}$, and the perturbation of its mirrored counterpart point $\hat{P}_i$, i.e., $\sigma_{\hat{P}_i} \cdot \overrightarrow{d_{\hat{P}_i}}$, across the symmetry plane $\mathcal{S}$,

we adjust their directions as follows:

$$\overrightarrow{d_{P_i}}^* = \text{Normalize}(\overrightarrow{d_{P_i}} + \alpha \cdot (\text{Mirror}(\overrightarrow{d_{\hat{P}_i}}, \mathcal{S}) - \overrightarrow{d_{P_i}})),$$
$$\overrightarrow{d_{\hat{P}_i}}^* = \text{Normalize}(\overrightarrow{d_{\hat{P}_i}} + \alpha \cdot (\text{Mirror}(\overrightarrow{d_{P_i}}, \mathcal{S}) - \overrightarrow{d_{\hat{P}_i}})), \tag{5}$$

where $\alpha$ is the step size for direction adjustment and Normalize() is an operation that normalizes a vector to unit length.

For the magnitude adjustment, we set them to be equal by taking the minimum between the two:

$$\sigma_{P_i} = \min(\sigma_{P_i}, \sigma_{\hat{P}_i}),$$
$$\sigma_{\hat{P}_i} = \min(\sigma_{P_i}, \sigma_{\hat{P}_i}). \tag{6}$$

Through the iterative execution of these processes, we successfully generate symmetric adversarial perturbations, thereby preserving the point cloud's symmetry to a certain extent. This preservation enables us to achieve a higher level of imperceptibility.

## 5 EXPERIMENTS

### 5.1 Experimental Setup

**Implementation.** We implement the SymAttack framework using PyTorch [21]. Semantic distances are calculated using features extracted and normalized from the penultimate layer of Point-M2AE [44]. We set the semantic distance threshold, $\tau_s$, at 0.001. The threshold for the number of points in a part, $\tau_P$, is set to 64 to prune excessively large parts, and the distance threshold for defining patches, $\tau_r$, is set at 0.1. For symmetry verification, the threshold $\tau_v$ is set to 0.01. The step size for direction adjustment, $\alpha$, is established at 0.5. All experiments are conducted on a workstation equipped with dual 2.40 GHz CPUs, 128 GB of RAM, and eight NVIDIA RTX 3090 GPUs.
**Datasets.** For our evaluation, we employ two renowned public datasets: ModelNet40 [37] and ShapeNet Part [40]. Within the ModelNet40 dataset, we allocate 9,843 point clouds for the training phase and 2,468 for testing. Similarly, for the ShapeNet Part dataset, 14,007 point clouds are designated for training, with 2,874 set aside for testing. To standardize the process, we uniformly sample 1,024 points from the surface of each object and scale them to fit within a unit cube, following the methodology described in [22].
**Victim DNN Classifiers.** We employ four well-established deep point cloud classifiers as victim models, including the multilayer perceptron (MLP)-based PointNet [22], the hierarchical neural network PointNet++ [23], the graph-based DGCNN [32], and the convolution-based PointConv [35]. These models are trained following the methodologies outlined in their respective original papers.
**Adversarial Attack Baselines.** To thoroughly validate the effectiveness of our method, we select six established adversarial attack methods as baselines. This selection encompasses gradient-based methods such as IFGM [3] and PGD [3], direction-based attacks like SI-Adv [9] and ITA [15], along with optimization-based strategies including GeoA$^3$ [33] and 3d-Adv [38].
**Evaluation Setting and Metrics.** To ensure a fair comparison, all attack methods are configured to achieve their highest possible attack success rate (ASR), which is the proportion of adversarial point clouds that effectively fool the DNN classifiers. In this maximal adversarialness setting [29], we assess the imperceptibility of the attacks. Specifically, we employ six established metrics to

**Table 1: Comparison on the perturbation sizes required by different methods to reach their highest achievable ASR in attacking PointNet, PointNet++, DGCNN and PointConv trained on ModelNet40 and ShapeNet Part.**

| Model | Attack | ModelNet40 | | | | | | | ShapeNet Part | | | | | |
|---|---|---|---|---|---|---|---|---|---|---|---|---|---|---|
| | | ASR (%) | CD ($10^{-4}$) | HD ($10^{-2}$) | $l_2$ | GR | Curv ($10^{-2}$) | EMD ($10^{-2}$) | ASR (%) | CD ($10^{-4}$) | HD ($10^{-2}$) | $l_2$ | GR | Curv ($10^{-2}$) | EMD ($10^{-2}$) |
| PointNet | PGD | 100 | 7.155 | 5.025 | 0.981 | 0.302 | 1.624 | 2.315 | 100 | 13.172 | 17.068 | 1.569 | 0.521 | 3.679 | 3.358 |
| | IFGM | 100 | 0.845 | 2.673 | 0.789 | 0.314 | 0.775 | 0.864 | 100 | 3.328 | 10.269 | 0.785 | 0.408 | 0.619 | 0.556 |
| | GeoA$^3$ | 100 | 4.646 | 0.497 | 1.307 | 0.121 | 0.396 | 2.319 | 100 | 7.531 | 1.444 | 2.655 | 0.146 | 0.465 | 4.101 |
| | 3d-Adv | 100 | 6.115 | 4.372 | 0.863 | 0.25 | 1.215 | 1.41 | 100 | 15.659 | 5.495 | 1.787 | 0.279 | 4.006 | 3.693 |
| | SI-Adv | 100 | 2.768 | 2.595 | 0.731 | 0.22 | 0.271 | 0.725 | 100 | 3.435 | 3.692 | 0.881 | 0.233 | 0.441 | 0.825 |
| | ITA | 100 | 2.747 | **0.414** | 0.534 | 0.122 | 0.555 | 1.214 | 100 | 5.872 | 1.917 | 1.002 | 0.181 | 1.016 | 2.035 |
| | Ours | 100 | **0.451** | 0.915 | **0.228** | **0.086** | **0.109** | **0.425** | 100 | **0.589** | **0.998** | **0.334** | **0.110** | **0.317** | **0.340** |
| PointNet++ | PGD | 100 | 5.182 | 0.636 | 0.753 | 0.125 | 1.508 | 2.146 | 100 | 10.09 | 3.257 | 1.342 | 0.215 | 3.969 | 3.328 |
| | IFGM | 100 | 3.558 | 1.162 | 0.64 | 0.146 | 1.149 | 1.454 | 100 | 4.532 | 3.608 | 0.548 | 0.22 | 1.824 | **1.584** |
| | GeoA$^3$ | 100 | 6.579 | **0.461** | 1.615 | 0.114 | 0.762 | 2.919 | 100 | 7.701 | 0.847 | 2.875 | 0.105 | 1.375 | 4.176 |
| | 3d-Adv | 100 | 8.915 | 3.564 | 1.535 | 0.141 | 1.288 | 2.784 | 100 | 9.564 | 3.778 | 2.014 | 0.197 | 3.021 | 3.59 |
| | SI-Adv | 100 | 9.399 | 2.377 | 1.422 | 0.185 | 1.061 | 2.684 | 100 | 9.266 | 3.233 | 1.535 | 0.203 | 1.146 | 2.811 |
| | ITA | 100 | 6.792 | 0.708 | 0.998 | 0.121 | 3.533 | 2.272 | 100 | 5.202 | **0.802** | 0.999 | 0.11 | 3.423 | 2.152 |
| | Ours | 100 | **0.801** | 1.023 | **0.381** | **0.073** | **0.328** | **1.159** | 100 | **2.357** | 1.435 | **0.517** | **0.092** | **0.768** | 1.671 |
| DGCNN | PGD | 100 | 19.968 | 5.098 | 1.933 | 0.267 | 4.924 | 4.785 | 100 | 63.556 | 27.557 | 5.224 | 0.511 | 7.275 | 9.233 |
| | IFGM | 100 | 15.791 | 12.391 | 1.622 | 0.363 | 2.849 | 3.777 | 100 | 19.623 | 26.04 | 2.069 | 0.504 | 4.954 | 4.387 |
| | GeoA$^3$ | 100 | 7.566 | 0.546 | 1.585 | 0.119 | 0.741 | 3.083 | 100 | 27.612 | 3.748 | 5.798 | 0.199 | **0.165** | 7.502 |
| | 3d-Adv | 100 | 10.345 | 3.807 | 3.589 | 0.227 | 5.997 | 6.685 | 100 | 21.553 | 8.531 | 2.258 | 0.282 | 5.119 | 4.628 |
| | SI-Adv | 100 | 7.146 | 1.691 | 1.087 | 0.143 | 0.666 | 2.495 | 100 | 11.685 | 3.019 | 1.772 | 0.160 | 5.054 | 3.646 |
| | ITA | 100 | 3.249 | **0.524** | 0.552 | 0.114 | 0.971 | 1.359 | 100 | 27.633 | 4.597 | 2.492 | 0.244 | 3.847 | 4.696 |
| | Ours | 100 | **1.346** | 0.761 | **0.407** | **0.079** | **0.305** | **1.144** | 100 | **4.356** | 1.202 | **0.312** | **0.134** | 0.217 | **1.908** |
| PointConv | PGD | 100 | 14.551 | 2.216 | 1.442 | 0.184 | 3.491 | 3.862 | 100 | 42.202 | 9.949 | 3.784 | 0.252 | 6.866 | 7.277 |
| | IFGM | 100 | 7.959 | 2.608 | 1.015 | 0.184 | 1.741 | 2.427 | 100 | 16.139 | 8.776 | 1.812 | 0.231 | 3.526 | 3.807 |
| | GeoA$^3$ | 100 | 6.809 | 0.644 | 2.169 | 0.119 | 1.119 | 3.556 | 100 | 9.383 | 1.222 | 4.224 | 0.12 | 1.19 | 5.391 |
| | 3d-Adv | 100 | 11.213 | 1.763 | 1.176 | 0.163 | 3.279 | 2.807 | 100 | 21.034 | 3.687 | 2.277 | 0.193 | 4.912 | 4.548 |
| | SI-Adv | 100 | 6.060 | 1.784 | 0.977 | 0.144 | 0.576 | 2.081 | 100 | 11.281 | 3.500 | 1.741 | 0.165 | 1.949 | 3.514 |
| | ITA | 100 | 5.539 | **0.480** | 0.833 | 0.111 | 1.904 | 1.971 | 100 | 3.082 | 1.452 | 1.375 | 0.146 | 3.654 | 2.925 |
| | Ours | 100 | **1.433** | 2.163 | **0.405** | **0.089** | **0.316** | **1.813** | 100 | **2.432** | **0.794** | **0.314** | **0.096** | **0.448** | 2.164 |

thoroughly measure distortions: Chamfer distance (CD) [5], Hausdorff distance (HD) [27], $l_2$-norm ($l_2$), curvature (Curv), geometric regularity (GR) [33], and earth mover's distance (EMD) [24].

## 5.2 Comparison with State-of-the-art Methods

**Attack and Imperceptibility Performance.** The results presented in Tab. 1 demonstrate that all methods can achieve a 100% attack success rate against the four DNN classifiers, but many of them introduce relatively large perturbations; for instance, PGD induces a distortion of approximately $7 \times 10^{-4}$ in terms of CD. By applying symmetry constraints, our method achieves the best performance across all datasets and DNN classifiers on five imperceptibility metrics and is predominantly optimal on the HD metric, thereby confirming the superiority of our approach.

**Visualization.** To illustrate how our approach enhances imperceptibility, we present visualizations of adversarial point clouds generated using various attack strategies on ModelNet40 aimed at fooling PointNet, as depicted in Fig. 3. Point clouds modified by PGD and IFGM exhibit noticeable outliers due to their less restrictive deformation techniques. In contrast, methods like GeoA$^3$, SI-Adv, and ITA, which leverage the geometric properties of shapes for modifications, result in significantly fewer visible outliers. Notably,

SymAttack, by preserving the symmetry of the shape, produces adversarial point clouds that are virtually free of outliers, thereby underscoring the effectiveness and superiority of our method in generating imperceptible attacks.

**Undefendability Performance.** We evaluate the robustness of SymAttack against various defense solutions, including simple random sampling (SRS), statistical outlier removal (SOR), denoiser and upsampler network (DUP-Net) [50], and IF-Defense [36]. The results, shown in Tab. 2, reveal that SymAttack keeps a success rate of over 99% against SRS, SOR, and DUP-Net defenses, and it still holds a 91% success rate against the strong IF-Defense. This performance validates SymAttack's effectiveness against defenses.

**Table 2: Comparison on ASR (%) of different methods in attacking PointNet with and without defense on ModelNet40.**

| Defense | IFGM | 3d-Adv | AdvPC | GeoA$^3$ | ITA | SI-Adv | Ours |
|---|---|---|---|---|---|---|---|
| - | 100.0 | 100.0 | 100.0 | 100.0 | 100.0 | 100.0 | 100.0 |
| SOR | 21.20 | 17.19 | 33.60 | 62.47 | 90.37 | 97.40 | 100.0 |
| SRS | 91.69 | 22.53 | 98.87 | 72.65 | 91.85 | 85.78 | 99.84 |
| DUP-Net | 16.29 | 12.30 | 29.00 | 73.70 | 85.41 | 95.80 | 99.80 |
| IF-Defense | 13.80 | 13.70 | 16.77 | 6.04 | 69.32 | 80.30 | 91.68 |

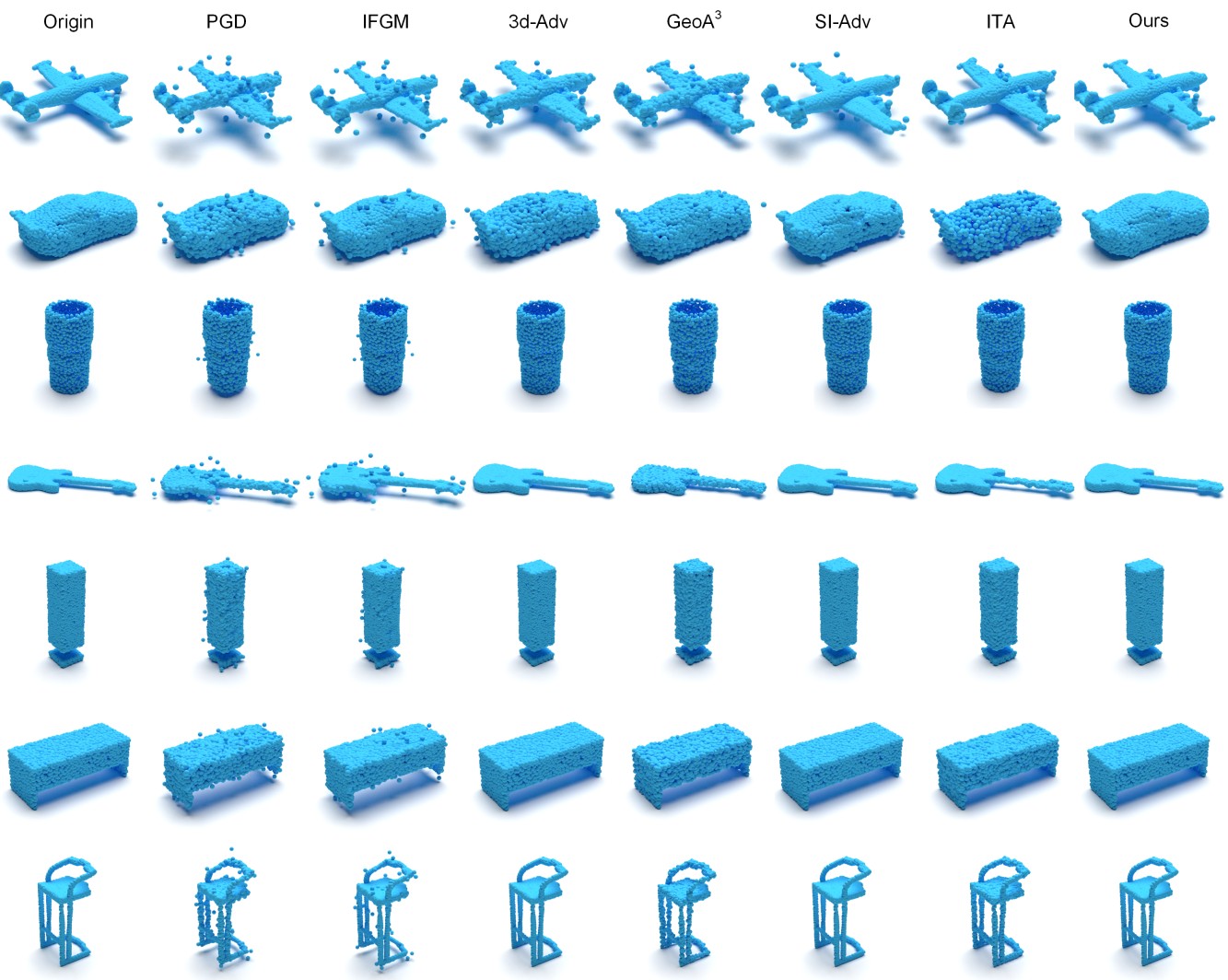

**Figure 3: Visualizations of original and adversarial point clouds generated to fool PointNet on the ModelNet40 dataset using different adversarial attack methods. The predicted categories before and after attack from top to bottom are:** AIRPLANE → PLANT; CAR → TENT; CUP → VASE; GUITAR → KEYBOARD; LAMP → WARDROBE; TABLE → DESK; CHAIR → STOOL.

## 5.3 Ablation Studies and Other Analysis

**Effects of Two Different Symmetric Elements.** To assess the significance of two types of symmetric elements, we compare the effectiveness of various SymAttack variants. Results in Tab. 3 indicate that removing any type of symmetric elements, i.e., part- and patch-level, degrades the imperceptibility of SymAttack, validating the importance of both types of elements. For a more intuitive understanding of the impact of removing one type of symmetric elements on the results, we also visualize the generated adversarial point clouds. As seen in Fig. 4, the absence of any one type of element leads to increased noise. Specifically, the lack of part-level elements has a more significant impact, underscoring the importance of maintaining symmetry in semantically consistent regions.

**Visualization of Symmetry Plane and Sampled Symmetric Elements.** To validate the efficacy of our method for detecting symmetric elements, we showcase visualizations of original point clouds, identified symmetry planes, and both part-level and patch-level symmetric elements in Fig. 5. Observations confirm that all symmetry planes are accurately detected and that critical object components are correctly identified as part-level symmetric elements, with patch-level symmetric elements being evenly distributed across the surface. Notably, objects lacking distinct components, such as LAMP, do not feature part-level symmetric elements. These findings robustly affirm the effectiveness of our symmetric element detection methodology, enhancing the implementation of symmetry-aware adjustments.

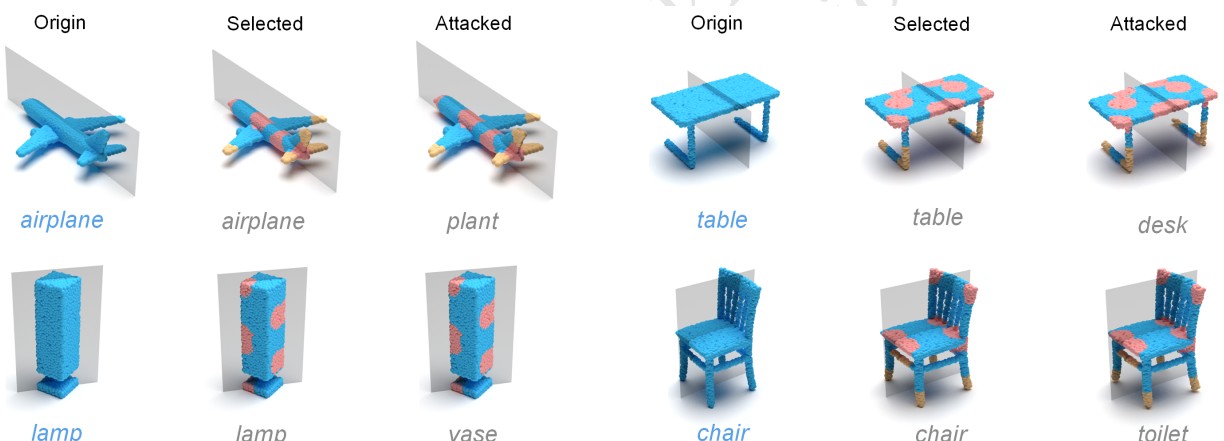

**Figure 4: Visualizations of original and adversarial point clouds generated by SymAttack and its variants, which exclude part-level and patch-level symmetry, to fool PointNet on the ModelNet40 dataset. Labels in blue represent the ground truth categories, while gray labels indicate the predicted categories post-attack.**

**Figure 5: Visualizations of original point clouds, selected part-level symmetric elements in yellow, patch-level symmetric elements in red, and adversarial point clouds generated by SymAttack to fool PointNet on the ModelNet40 dataset. Labels in blue represent the ground truth categories, while gray labels indicate the predicted categories post-attack.**

**Table 3: Comparison on the attack and imperceptibility performance of SymAttack with and without considering part- and patch-level symmetry.**

| Model | Attack | ModelNet40 | | | | | | | ShapeNet Part | | | | | |
|---|---|---|---|---|---|---|---|---|---|---|---|---|---|---|
| | | ASR (%) | CD ($10^{-4}$) | HD ($10^{-2}$) | $l_2$ | GR | Curv ($10^{-2}$) | EMD ($10^{-2}$) | ASR (%) | CD ($10^{-4}$) | HD ($10^{-2}$) | $l_2$ | GR | Curv ($10^{-2}$) | EMD ($10^{-2}$) |
| PointNet | Ours w/o part | 100 | 1.073 | 2.465 | 0.662 | 0.234 | 0.325 | 0.992 | 100 | 1.321 | 2.411 | 1.013 | 0.282 | 0.735 | 0.794 |
| | Ours w/o patch | 100 | 0.576 | 1.297 | 0.409 | 0.173 | 0.284 | 0.675 | 100 | 0.925 | 1.623 | 0.531 | 0.247 | 0.497 | 0.694 |
| | Ours | 100 | **0.451** | **0.915** | **0.228** | **0.086** | **0.109** | **0.425** | 100 | **0.589** | **0.998** | **0.334** | **0.110** | **0.317** | **0.340** |

**Effects of Different Perturbation Magnitude Settings.** We investigate the effect of different perturbation magnitude settings on the performance of SymAttack during the symmetry-aware adjustment process. The settings explored include maintaining the original magnitude and adjusting the magnitudes on two symmetric points to either the larger or smaller values between them. The results, presented in Tab. 4, demonstrate that these variations in

**Table 4: Comparison on attack and imperceptibility performance of SymAttack under different perturbation magnitude settings. "Ours (N)" indicates no change to the original perturbation magnitude, while "Ours (L/S)" denotes adjusting the perturbation magnitude of two symmetric points to either the larger or smaller of the two.**

| Model | Attack | ModelNet40 | | | | | | | ShapeNet Part | | | | | |
| | | ASR (%) | CD ($10^{-4}$) | HD ($10^{-2}$) | $l_2$ | GR | Curv ($10^{-2}$) | EMD ($10^{-2}$) | ASR (%) | CD ($10^{-4}$) | HD ($10^{-2}$) | $l_2$ | GR | Curv ($10^{-2}$) | EMD ($10^{-2}$) |
|---|---|---|---|---|---|---|---|---|---|---|---|---|---|---|---|
| PointNet | Ours (N) | 100 | 0.451 | **0.915** | 0.228 | 0.086 | 0.109 | 0.425 | 100 | 0.589 | 0.998 | 0.334 | 0.110 | 0.317 | 0.340 |
| | Ours (L) | 100 | 0.464 | 1.113 | 0.238 | 0.102 | 0.117 | 0.454 | 100 | 0.611 | 1.107 | 0.356 | 0.121 | 0.384 | 0.394 |
| | Ours (S) | 100 | **0.445** | 0.927 | **0.226** | **0.085** | **0.097** | **0.411** | 100 | **0.553** | **0.998** | **0.316** | **0.106** | **0.312** | **0.327** |

**Table 5: Comparison on attack and imperceptibility performance of different methods, both with and without applying our symmetry-aware adjustment in attacking PointNet on ModelNet40 and ShapeNet Part.**

| Model | Attack | ModelNet40 | | | | | | | ShapeNet Part | | | | | |
| | | ASR (%) | CD ($10^{-4}$) | HD ($10^{-2}$) | $l_2$ | GR | Curv ($10^{-2}$) | EMD ($10^{-2}$) | ASR (%) | CD ($10^{-4}$) | HD ($10^{-2}$) | $l_2$ | GR | Curv ($10^{-2}$) | EMD ($10^{-2}$) |
|---|---|---|---|---|---|---|---|---|---|---|---|---|---|---|---|
| PointNet | IFGM | 100 | 0.845 | 2.673 | 0.789 | 0.314 | 0.775 | 0.864 | 100 | 3.328 | 10.269 | 0.785 | 0.408 | 0.619 | 0.556 |
| | Sym-IFGM | 100 | **0.796** | **1.71** | **0.367** | **0.148** | **0.164** | **0.449** | 100 | **1.372** | **2.530** | **0.689** | **0.297** | **0.602** | **0.441** |
| | GeoA$^3$ | 100 | 4.646 | 0.497 | 1.307 | 0.121 | 0.396 | 2.319 | 100 | 7.531 | 1.444 | 2.655 | 0.146 | 0.465 | **4.101** |
| | Sym-GeoA$^3$ | 100 | **3.653** | **0.488** | **1.159** | **0.117** | **0.289** | **1.918** | 100 | **3.332** | **1.163** | **2.241** | **0.140** | **0.333** | 4.124 |
| | ITA | 100 | 2.747 | **0.414** | 0.534 | **0.122** | 0.555 | 1.214 | 100 | 5.872 | 1.917 | 1.002 | **0.181** | 1.016 | 2.035 |
| | Sym-ITA | 100 | **1.314** | 0.682 | **0.457** | 0.124 | **0.249** | **0.341** | 100 | **3.736** | **1.678** | **0.882** | 0.185 | **0.762** | **1.254** |
| | SI-Adv | 100 | 2.768 | 2.595 | 0.731 | 0.22 | 0.271 | 0.725 | 100 | 3.435 | 3.692 | 0.881 | 0.233 | 0.441 | 0.825 |
| | Sym-SI-Adv | 100 | **0.563** | **1.13** | **0.284** | **0.104** | **0.133** | **0.519** | 100 | **0.717** | **1.246** | **0.416** | **0.135** | **0.381** | **0.415** |

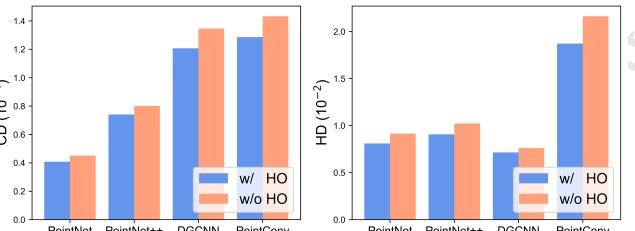

**Figure 6: Imperceptibility of SymAttack measured by CD and HD in attacking four DNN classifiers on ModelNet40, with and without considering high-order symmetry (HO).**

magnitude settings have minimal impact on the outcomes. However, setting the perturbation magnitudes on symmetric points to the smaller of the two values consistently yields better results. Therefore, we adopt this setting in our methodology.

**Analysis on High-order Symmetry.** Many objects, particularly parts within an object such as the four wheels of a car, possess more than one plane of symmetry. Thus, we delve into second-order symmetry, entailing the utilization of two symmetry planes. For each identified symmetric element, we conduct a symmetry-aware adjustment individually for both symmetry planes. The results showcased in Fig. 6 reveal that integrating high-order symmetry indeed leads to a modest enhancement in the imperceptibility of attacks across all four DNN classifiers, although the degree of improvement is somewhat restrained. As a result, we opt to utilize a single plane of symmetry in this paper.

**Generalization Ability of Symmetry Perseaving.** To evaluate the generalizability of our symmetry-preserving strategy, we incorporate it into four established iterative adversarial attack methods:

IFGM, GeoA$^3$, SI-Adv, and ITA. Results, as shown in Tab. 5, indicate that these methods, augmented with our symmetric elements detection and symmetry-aware adjustment, exhibit significant improvements across most performance metrics under identical parameter settings. These outcomes substantiate the broad applicability and effectiveness of our symmetry-preserving approach.

**Analysis on Asymmetric Cases.** We also investigate the efficacy of SymAttack in attacking objects that are not perfectly symmetrical. The visualizations of TABLE and GUITAR in the bottom row of Fig. 4 indicate that, compared to entirely symmetrical cases, adversarial point clouds generated from asymmetrical objects contain a small number of noise points. Aside from this, the imperceptibility of the generated adversarial point clouds remains impressively high. This demonstrates a certain level of robustness in our method when dealing with asymmetric cases to some extent.

## 6 CONCLUSION

In this paper, we have proposed SymAttack, a novel framework for imperceptible adversarial attacks on 3D point clouds. The rationale involves identifying a subset of representative symmetric elements and then adjusting the perturbations on points located within symmetric elements to maintain symmetry. Extensive experiments validate that SymAttack successfully generates adversarial point clouds with significantly enhanced imperceptibility.

**Limitation and Future Work.** Our method assumes that the object's shape is generally symmetrical, which means it may not work well with severely asymmetrical shapes. In the future, we aim to develop a method for identifying local symmetries and apply our approach to these locally symmetrical regions.

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
