# OpenReview forum: "SymAttack: Symmetry-aware Imperceptible Adversarial Attacks on 3D Point Clouds"
_acmmm.org/ACMMM/2024/Conference — MM2024 Poster_

### Official Review · Reviewer_8CDR · 2024-05-21

**Rating:** 4
**Confidence:** 3

**Summary:**

In this paper, the authors attribute the inadequate imperceptibility of adversarial attacks on 3D point clouds to significant deviations from symmetry. To address this issue, they propose a novel framework, symmetry-aware imperceptible adversarial attacks on 3D point clouds (SymAttack). This framework identifies part- and patch-level symmetry elements and adjusts perturbations relative to the symmetry plane during the attack iterations. By preserving symmetry within the attack process, SymAttack enhances the imperceptibility of the adversarial attack strategy.

**Strengths:**

1. This paper proposes that focusing on symmetry during adversarial attacks on 3D point clouds can lead to more imperceptible adversarial examples.
2. The paper includes comprehensive experiments on multiple DNN classifiers and datasets, which are helpful in verifying the effectiveness of the proposed method.
3. The paper is well-organized and clearly written, with detailed explanations of the methodology and well-presented results, including visualizations and quantitative comparisons.

**Limitations:**

1.When the adversarial disturbance is added to irregular objects, the proposed method may fail, which reflects the scene restriction and lack of generalization of the proposed method, as acknowledged by the authors.
2. There is a lack of experiments using black-box settings in the paper. A broader range of comparisons could strengthen the evaluation.

**Suitability:**

2

---

### Official Review · Reviewer_WzNL · 2024-05-23

**Rating:** 4
**Confidence:** 4

**Summary:**

This paper introduces a framework for generating imperceptible adversarial attacks on 3D point clouds by preserving the inherent symmetry of the objects. The authors argue that existing adversarial attacks often disrupt the symmetry, making them perceptible. The proposed method, SymAttack, identifies symmetric elements at part and patch levels and adjusts perturbations relative to the symmetry plane during attack iterations. The paper validates the effectiveness of SymAttack through extensive experiments, demonstrating its superiority over state-of-the-art methods in terms of imperceptibility and robustness against various defenses.

**Strengths:**

1. The authors provide a novel perspective by attributing the perceptibility of attacks to the disruption of symmetry in the generated point clouds.
2. The proposed SymAttack framework is well-motivated and technically sound. The authors present a clear methodology for identifying symmetric elements at both part and patch levels, and adjusting perturbations to maintain symmetry. The symmetric element detection process is illustrated with visualizations, enhancing the clarity of the approach.
3. The authors conduct thorough ablation studies to analyze the effects of different components in SymAttack, such as the importance of part-level and patch-level symmetric elements, different perturbation magnitude settings, and the impact of high-order symmetry. These studies provide valuable insights into the proposed method.

**Limitations:**

1. The authors mention SymAttack has some robustness in handling asymmetric objects.  The method assumes that the objects have a significant degree of symmetry. While this is true for many 3D objects, there are notable exceptions. The authors mention this limitation and suggest exploring local symmetries as future work, but providing some initial results or discussions on this aspect could strengthen the paper.
2. The process of identifying and verifying symmetric elements, especially at part and patch levels, could be computationally intensive. The paper should discuss the scalability of the approach for large point clouds and complex objects.
3. The paper does not provide a clear justification for the choice of threshold values used in the symmetric element detection process ($\tau_s$, $\tau_P$, $\tau_r$, and $\tau_v$). A sensitivity analysis of these thresholds or a discussion on how they were determined would improve the reproducibility and robustness of the proposed method.

**Suitability:**

3

---

### Official Review · Reviewer_SPf4 · 2024-05-27

**Rating:** 5
**Confidence:** 3

**Summary:**

This paper points out that existing point cloud attack methods still suffer from the issue of poor imperceptibility. The primary reason is that the human eye is adept at detecting inconsistencies in regular structures, and adversarial attack methods often disrupt the structural regularity of the point cloud, especially its symmetry. Therefore, the authors propose SymAttack, which first identifies the symmetrical parts of the point cloud. During the optimization process, it iteratively ensures that the adversarial perturbation vectors of the symmetrical parts are symmetrical in direction and equal in magnitude, thereby improving the imperceptibility of the adversarial point cloud.

**Strengths:**

(1)	The idea of enhancing imperceptibility based on symmetry is novel and well-motivated. It sounds reasonable.
(2)	The overall logic of the article is coherent, the writing is well-crafted, and the authors' viewpoints are clearly expressed.
(3)	Extensive experiments and visualization results demonstrate the effectiveness of SymAttack, and the results appear to be easily reproducible.

**Limitations:**

(1)	The example in Figure 1 is not typical enough, as the adversarial point clouds in (b) and (c) do not show significant differences and both appear quite imperceptible. This does not strongly demonstrate the detrimental effect of asymmetry on imperceptibility.
(2)	The data in Table 1 appears somewhat strange. Why does IFGM perform so much better than PGD in the attack experiments on PointNet and PointNet++, even significantly outperforming other imperceptible point cloud attack methods?
(3)	I am curious whether SymAttack can maintain its adversarial effectiveness after being subjected to benign resampling preprocessing defense methods, as this would determine the practical significance of the adversarial samples in the physical world.
(4)	Additionally, how would the attack effectiveness change if different axes of symmetry were chosen?

**Suitability:**

2

---

### Meta-Review · Area_Chair_iokT · 2024-06-27

**Recommendation:** Accept (Poster)
**Confidence:** 4

**Metareview:**

Authors please follow comments to comprehensively revise the paper, including issues like method clarity, experiments enrichment etc.